# Human Airway Epithelium Responses to Invasive Fungal Infections: A Critical Partner in Innate Immunity

**DOI:** 10.3390/jof9010040

**Published:** 2022-12-27

**Authors:** Arianne J. Crossen, Rebecca A. Ward, Jennifer L. Reedy, Manalee V. Surve, Bruce S. Klein, Jayaraj Rajagopal, Jatin M. Vyas

**Affiliations:** 1Division of Infectious Diseases, Massachusetts General Hospital, Boston, MA 02114, USA; 2Department of Medicine, Harvard Medical School, Boston, MA 02115, USA; 3Center for Regenerative Medicine, Massachusetts General Hospital, Boston, MA 02114, USA; 4Department of Pediatrics, School of Medicine and Public Health, University of Wisconsin-Madison, Madison, WI 53706, USA; 5Department of Medicine, School of Medicine and Public Health, University of Wisconsin-Madison, Madison, WI 53706, USA; 6Department of Medical Microbiology and Immunology, School of Medicine and Public Health, University of Wisconsin-Madison, Madison, WI 53706, USA; 7Division of Pulmonary and Critical Care Medicine, Department of Medicine, Massachusetts General Hospital, Boston, MA 02114, USA; 8Harvard Stem Cell Institute, Cambridge, MA 02138, USA; 9Klarman Cell Observatory, Broad Institute of Massachusetts Institute of Technology and Harvard, Cambridge, MA 02142, USA

**Keywords:** human airway epithelium, pulmonary fungal infections, host responses, air-liquid interface, in vitro human airway epithelial models, *Aspergillus fumigatus*, endemic fungi

## Abstract

The lung epithelial lining serves as the primary barrier to inhaled environmental toxins, allergens, and invading pathogens. Pulmonary fungal infections are devastating and carry high mortality rates, particularly in those with compromised immune systems. While opportunistic fungi infect primarily immunocompromised individuals, endemic fungi cause disease in immune competent and compromised individuals. Unfortunately, in the case of inhaled fungal pathogens, the airway epithelial host response is vastly understudied. Furthering our lack of understanding, very few studies utilize primary human models displaying pseudostratified layers of various epithelial cell types at air-liquid interface. In this review, we focus on the diversity of the human airway epithelium and discuss the advantages and disadvantages of oncological cell lines, immortalized epithelial cells, and primary epithelial cell models. Additionally, the responses by human respiratory epithelial cells to invading fungal pathogens will be explored. Future investigations leveraging current human in vitro model systems will enable identification of the critical pathways that will inform the development of novel vaccines and therapeutics for pulmonary fungal infections.

## 1. Should We Care about Pulmonary Lung Infections?

Pulmonary fungal infections result in high mortality rates, especially in patients with compromised immune status, because of insensitive diagnostics, lack of molecular understanding of the pathophysiology of invasive fungal infections (IFIs), and limited knowledge of the nature of coordination between epithelial cells and the immune system to develop a protective response. This highlights the critical need to better understand host defense against invading fungi. In the United States, direct healthcare costs from hospitalizations and outpatient visits due to fungal infections amount to $7.3 billion annually, with more than $2.4 billion specifically associated with the treatment of pulmonary fungal pathogens [1]. This estimate is likely significantly lower than actual healthcare costs due to the underdiagnosis of IFIs. The primary pathogens responsible for IFIs in the lungs include *Aspergillus*, *Cryptococcus*, *Pneumocystis*, *Coccidioides*, *Paracoccidioides*, *Blastomyces*, and *Histoplasma* (Table 1) [2]. These fungal organisms are typically classified into two categories: opportunistic and endemic/dimorphic.

While treatments for cancer, end-organ failure, and autoimmune disease have improved over the past decade, these immunomodulatory strategies also increase the risk of opportunistic fungal infections. Indeed, opportunistic fungal infections exploit high-risk patient populations, including those with cancer, solid organ or hematopoietic stem cell transplantation, autoimmune disease, HIV infection, and pre-existing respiratory diseases (e.g., asthma, cystic fibrosis [CF], chronic obstructive pulmonary disease [COPD]) [20,21]. Individuals with prior pulmonary viral infections (e.g., influenza and SARS-CoV-2) also experience elevated risk of pulmonary fungal superinfections [22,23,24,25,26,27], although the mechanism of increased susceptibility remains poorly understood. These patient populations have steadily risen over the past ten years, correlating with increased IFI prevalence and mortality [26]. *Aspergillus*, *Cryptococcus*, and *Pneumocystis* are the most prominent opportunistic fungal pathogens in the lungs [7,20,28].

Contrary to opportunistic pathogenic fungi that largely impact immunocompromised individuals, endemic fungi can cause a range of disease severity regardless of their host’s immune status (i.e., immunocompetent or immunocompromised). Although these organisms are typically restricted to a specific geographic region, current reports suggest these fungi are expanding well beyond their historical areas, in some cases likely due to changing climate, thus amplifying the risk of acquiring fungal disease burden [29]. The most prominent endemic fungi contributing to pulmonary fungal infections in humans include *Coccidioides*, *Paracoccidioides*, *Blastomyces*, and *Histoplasma* [29]. Although these pathogens infect individuals regardless of immune status, an immunocompromised status amplifies the risk of severe disease and mortality (Table 1). Furthermore, classically opportunistic fungal infections, such as cryptococcosis, have also been identified in immunocompetent individuals (i.e., *C. gattii* infections), though far less frequently than their dimorphic counterparts [30,31,32].

Clinical data and current literature indicate that several components of the immune system are critical for the swift clearance of fungal pathogens. Still, the rules that govern inflammatory responses to fungi in the lungs are not well understood. Past studies have focused on innate and adaptive immune responses but have overlooked a key player in these respiratory infections: the airway epithelium. Here, we review in vitro models utilized to study the human airway epithelium, the complexity of the respiratory epithelium in humans, and its contribution to host responses to invasive fungal pathogens. While the respiratory epithelium spans the upper (nasal cavity, pharynx, larynx) and lower (conducting airways and respiratory zone) respiratory tract, this review will focus on the lower respiratory tract, with particular focus on the bronchioles (lower airways) and alveolar regions.

## 2. Viewing Human Airway Epithelium as a Monomorphic Barrier Is Outdated

Investigations have revealed that the respiratory epithelium is a complex, immunologically active cell network composed of distinct cell types. Single-cell sequencing of lung tissue and bronchoalveolar lavage samples unveiled the vast complexity of the human airway epithelium [33,34]. Once thought to be comprised primarily of ciliated cells, club cells, goblet cells, and basal cells, the small airway epithelium is pseudostratified with both common cell types (basal, ciliated, club, goblet) and rare cell types (ionocytes, neuroendocrine, tuft cells), each with specialized functions [34,35,36].

Basal cells are multipotent stem cells critical for the repair and regeneration of the epithelium and account for a third of epithelial cells in the respiratory tract [37]. As the name suggests, ciliated cells promote the clearance of mucus and debris through coordinated beating of cilia pods. Secretion of mucus, as well as peptides with anti-microbial and anti-inflammatory properties, is mediated through the two secretory cells in epithelium—club and goblet cells. Recently identified ionocytes regulate ion transport, fluid, and pH [35,38,39]. Additionally, this rare cell type is the primary expressor of the cystic fibrosis transmembrane conductance regulator (CFTR). Neuroendocrine cells in the airway epithelium act as environmental sensors and send signals to the central nervous system [40]. Functioning as chemosensory cells, tuft cells have been linked to type 2 immunity and are thought to synthesize cysteinyl leukotrienes critical to the coordinated influx of immune cells [41]. Another cell type recently described is the microfold (M) cell, which can sample the environment, translocate pathogens, and is often associated with underlying patches of immune cell follicles termed bronchus-associated lymphoid tissue (BALT) [42,43,44].

Upon inhalation of a pathogen, the airway epithelium is the first point of contact, suggesting these cells may be critical to shaping proper host responses. Indeed, responses by the epithelium trigger mucous production, coordinated cilia action, and recruitment of immune cells to the site of the pathogen [45,46]. Respiratory epithelial cells can sense and respond to inhaled pathogens by activating pattern recognition receptors (PRRs), leading to the secretion of pro-inflammatory chemokines, and subsequent recruitment of innate immune cells [47]. Below, we will introduce models utilized to study human airway epithelium. While there are resident immune cells in the lungs (i.e., alveolar macrophages, tissue-resident memory T cells [T_RM_]), these are beyond the scope of this review.

### 2.1. Mouse Models and Cell Lines Do Not Phenocopy Human Respiratory Epithelial Biology

Murine pulmonary fungal infections models are beneficial for dissecting the molecular pathways involved in the response to fungal infections and examining the impact of novel therapeutic targets on infection progression and mortality rates. Still, they do not fully recapitulate human disease. As an example, immunocompetent mice are highly resistant to *Aspergillus fumigatus* delivered intratracheally. Indeed, most models require high dose steroids to recapitulate lung phenotypes [48]. Another example is highlighted by the failure of the formalin killed spherule vaccine to protect humans in clinical trials, although the vaccine was highly protective in murine models against *Coccidioides* infection [49]. Human in vitro models are limited by challenges in cell culture techniques. Nearly all published studies of human airway epithelium utilize cell lines, including the cancer lines A549, H292, and Calu-3 [50,51,52], as well as immortalized non-cancerous bronchial epithelial lines (BEAS-2B, HBEC3-KT, and 16HBE) (Table 2). A549 is the most widely utilized cell line to study fungal infections. While these easily passaged and long-lasting cell lines have provided valuable data to further our understanding of the respiratory epithelium, numerous limitations reduce the impact of these studies alone.

Human cancer cell lines (i.e., A549, Calu-3, H292) have monomorphic oncogenic nature that does not recapitulate the fully differentiated, pseudostratified characteristics of primary human airway epithelium. The simplification of the diverse human airway epithelium observed in these cell lines removes highly specialized networks seen in vivo in disease and infections. Emerging data indicate that specialized cell types in primary airway epithelium contribute differentially to the integrated response. Hence, monomorphic cell lines lack the ability to participate in complex intercellular communications. For example, these cell lines lack club cells, an epithelial cell type critical in driving *Aspergillus*-mediated allergic inflammation [53]. Another epithelial subtype lacking in these cell lines is ionocytes, which are the primary CFTR producers and essential to CF pathophysiology. CFTR is, however, present in Calu-3 cells [54]. Although ciliated cells are abundant and critical to epithelial function, these cancer cell lines are deficient in beating cilia. Rather than forming the pseudostratified layer with multiple cell subtypes seen in tissue, these lung epithelial cell lines form a monolayer. Furthermore, A549 and H292 cells must be grown submerged in media, unlike primary cells, which grow in an air-liquid interface (ALI) as seen in vivo. Calu-3 cells grown at ALI form pseudostratified layers rather than the monolayer observed in submerged culture [55]. Although most similar of the cell lines to primary epithelial cells, Calu-3 has not been widely utilized in fungal infection models [56,57].

**Table 2 jof-09-00040-t002:** Human respiratory epithelial cell lines.

Cell Line	Source	Immortalization Method	Sex & Ethnicity of Source	Ref
A549	Explant culture of lung carcinomatous tissue; alveolar	N/A	Male; Caucasian	[58]
BEAS-2B	Normal human bronchial epithelium (non-cancerous individuals)	Adenovirus 12-SV40 virus	Male; No ethnicity noted	[59,60]
Calu-3	Pleural effusion from an individual with lung adenocarcinoma; bronchial	N/A	Male; Caucasian	[61]
HBEC3-KT	Primary human bronchial epithelial cell	hTERT expression & mouse CDK4-expressing retrovirus	Female; No ethnicity noted	[62]
H292	Lymph node metastasis of a pulmonary mucoepidermoid carcinoma	N/A	Female; Black	[63]
16HBE	Human bronchial epithelial cell from a heart-lung patient	Origin-of-replication defective SV40 plasmid	Male; No ethnicity noted	[64]

Although the immortalized cells derived from non-tumorigenic lung tissue (BEAS-2B, 16HBE, and HBEC3-KT) are not oncogenic, these cells also have significant limitations. Many of these cell lines undergo squamous differentiation, suggesting a uniformity not seen in the human body. Culture conditions (i.e., presence of fetal bovine serum [FBS]) mediate this differentiation in BEAS-2B cells, leading to varied sensitivity to stimulation and altered gene expression and cell metabolism [65]. The presence of FBS may explain the lack of differentiation into multiple sub-types seen in one study, which observed mucin 5AC (MUC5AC)-positive goblet cells in primary epithelium but not BEAS-2B cells [66]. The BEAS-2B line exhibits similar characteristics to mesenchymal stem cells, including surface markers, suppressive activity, and osteogenic and adipogenic differentiation [67]. Immortalized with SV40 plasmid, 16HBEs have been widely used to investigate asthma, COPD, and cancer. The advantages of 16HBE cells include the expression of mucus on the epithelium [68], expression of the critical CF channel, CFTR, and has been noted to maintain many morphological phenotypes and functions seen in primary cells [64,68]. Unfortunately, these cells lack cilia, a significant structure for host defense by epithelial cells. Furthermore, these cells grow as a mix of monolayer and multilayer. Comparisons of immortalized and primary cells have suggested that primary epithelial cells are more sensitive to toxins (e.g., mycotoxin from *Fusarium*) [69].

Of the cell lines originating from non-cancer patients, HBEC3-KT is most similar to primary cells. Indeed, HBEC3-KT is grown in pseudostratified layers under ALI conditions and produces mucus-secreting cells and cilia [70]. The existence of rare cell types in this system has not been reported. Although these cells are closest to primary cultures, IL-6 secretion and mucociliary differentiation are also impaired compared to primary airway epithelium [71]. Furthermore, this line lacks inflammatory characteristics such as the absence of LPS-induced secretion of pro-inflammatory cytokines (e.g., tumor necrosis factor α [TNFα], interleukin [IL]-1β, IL-12) [72]. While the ease of passaging and growing oncogenic and immortalized cells has led to these lines being widely utilized, they lack key features of primary epithelium, limiting the ability to make generalized conclusions using this system. Hence, primary cells are critical to understanding human biology.

### 2.2. Towards a Better Model of Human Airway Epithelium

Primary human airway epithelial cells (hAECs) derived from epithelial stem cells circumvent the limitations of immortalized and oncologic cell lines. Unfortunately, primary cells grow slowly, can be more technically challenging, and are limited in the number of times they can be passaged. Co-culture with fibroblasts and Rho-associated protein kinase (ROCK) inhibition improved the use of epithelial stem cells, but the lifespan remained short [73,74]. Recently, a novel method for extended culturing of patient-derived hAECs was created through dual SMAD inhibition in basal cells [75]. Basal cells from patient sputum, bronchoalveolar lavage (BAL) fluid, or tissue samples can be isolated, expanded beyond ten passages, and differentiated into airway epithelium. These cultures recapitulate the true diversity of conducting airway epithelium with common and rare cell types in pseudostratified layers, enabling mechanistic studies in rare populations and the epithelial network in disease [35]. Additionally, these cells can be grown at ALI, allowing the formation of polarized barriers and investigation of basolateral and apical responses in disease.

While the primary hAECs ALI model may not fully recapitulate the in vivo environment (e.g., lacks resident immune cells, endothelial cells, fibroblasts), this model offers the advantages of a highly pliable system to conduct mechanistic studies without compromising relevant human physiology. A key benefit is the ability to isolate basal cells from specific patient populations to better understand the epithelial contribution to pathophysiology. Indeed, these differentiated primary hAECs have been isolated from healthy people as well as patients with CF, COPD, and asthma [76,77,78]. While promising, these studies have not been expanded to those patients with known prior IFIs to determine if epithelial cells retain the memory of prior infection (e.g., epigenetic modifications, altered cytokine profile). It should be noted that substantial variability in differentiated epithelium likely exists between samples collected from multiple individuals. Thus, pertinent information should be tracked and considered during data analyses, including age, sex, ethnicity, underlying lung disease, and known medications. Furthermore, expanding studies to include co-culture with primary hAEC and relevant immune cells provides an ideal platform to dissect lung immune responses. Leveraging the ALI hAECs to investigate host defense against invading pathogens, particularly from high-risk populations (e.g., CF, COPD, cancer), is an essential tool for understanding human lung biology.

## 3. The Human Airway Epithelium Is a Key Player in the Host-Fungal Pathogen Battle

Much of our current understanding of pulmonary fungal infections comes from animal models, which likely does not fully recapitulate human diseases. While much of these data are important to the field, we will focus on studies using human airway epithelial models. Narrowing the focus to human airway epithelial studies drastically reduced the literature search for lung epithelium responses to fungal pathogens, highlighting this critical gap in knowledge. Our understanding of airway epithelial responses to endemic fungi is minimal. Responses of airway epithelium to *Coccidioides* have not been reported in human or animal models. Investigations into *Blastomyces, Histoplasma*, and *Paracoccidioides* primarily utilize experimental models rather than human airway epithelial cell systems [79,80]. Opportunistic fungal infections, especially *Aspergillus* spp., and human airway epithelial cells are more commonly studied compared to endemic fungi, but limited information is available for *Cryptococcus*, *Mucor*, *Rhizopus*, and *Pneumocystis*. Thus, we mainly focus on the epithelial-mediated immune response to invasive pulmonary aspergillosis (IPA). This section will flow from the initial inhalation of a pathogen to airway defenses and then to downstream recruitment of immune cells to the site of infection. In addition to discussing immunity driven by human airway epithelium, we briefly examined models for bacterial and viral pulmonary infections in primary cells that can be adapted to the field of fungal immunology.

### 3.1. First Contact with Fungal Pathogens

The conducting airway epithelium is the first point of contact for inhaled fungal pathogens. As the first line of defense, the airway epithelium, the associated mucus, and resident alveolar macrophages are critical for removing conidia. In addition to the barrier function conferred by lung epithelium, immune cells are vital to the containment and clearance of fungi. Upon disruption, the fungal conidia and spores are inhaled and can reach the distal airways in the lungs. Studies using human biopsy samples and cadaver tissue demonstrate that *Aspergillus* infections in the lung are a disease of the distal airways and not the alveolar spaces [81]. Notably, the proximal airways remove inhaled *Aspergillus* conidia through mucociliary clearance, but impairment of this response enables colonization and invasion [82]. *Coccidioides* arthroconidia can reach the terminal airways [83], but the contribution of the epithelium in response remains poorly understood for this and other endemic fungi. Models of *Pseudomonas aeruginosa* bacterial infections revealed that mucin 1 (MUC1) mediated bacterial adherence, dampened inflammation, and facilitated infection in airway epithelium [84,85]. MUC1 also binds to *Pneumocystis* and contributes to decreased inflammatory signaling [86], suggesting fungal organisms may leverage host-derived mucins to facilitate invasion.

Prior to interactions directly with lung epithelium, fungal organisms secrete proteases and cytolytic toxins that can disrupt barriers through tight junction break down, facilitate epithelial remodeling, and trigger immune responses. In *Aspergillus* infections, proteases play a role in fungal asthma and allergic fungal airway disease. Alkaline protease 1 (Alp1) levels from *Aspergillus* correlate with asthma and disease severity in patients [87], which elicit eosinophilic infiltration in the lungs and damage cell junctions [53]. Other protease implicated in disease and airway remodeling from fungal pathogens include matrix metalloprotease (Asp f 5) and serine protease (Asp f 13) [88]. Furthermore, protease-activated receptor-2 (PAR-2) on human epithelium, which recognizes fungal proteases, skews immune responses towards a type 2 bias in allergic diseases [89]. Little is known about the role of fungal-derived protease in IPA. In addition to proteases, fungal pathogens can release toxins that disrupt host barriers. Gliotoxin, a mycotoxin derived from *Aspergillus*, causes airway epithelium disruption through destabilization of actin filaments [90]. Proteases and cytolytic toxins may promote invasion through the break down of the lung epithelial barrier and promote immune responses and warrants further investigations in IFIs.

While fungal pathogens employ mechanisms to disrupt host defenses, the airway epithelium maintains the critical barrier essential for the resistance to fungal infections. One mechanism of protection for this barrier occurs through the production of antimicrobial proteins and peptides (AMPs; e.g., lysozymes, β-defensin, ferrins) by lung epithelial cells. The mucus layer on the epithelial barrier contains these AMPs, which can provide host defense prior to direct interaction with the inhaled pathogen. Lysozyme enables damage to polysaccharides on microbes and decreases metabolic activity of *A. fumigatus* hyphae and *Histoplasma* [91]. Airway epithelial cells increase β-defensin-1 in response to *A. fumigatus* and restricts grown for germinating conidia, but not hyphae [91,92]. Transferrin and lactoferrin deplete iron from the environment and blunt microbial growth. Like β-defensin, lactoferrin inhibits gemination of conidia, but not hyphal growth, revealing that gemination may require more iron [91]. Mucormycosis amplifies serum iron levels, which indicates a potential deficit in lactoferrin or transferrin function [93]. These AMPs could be targeted as a therapeutic strategy. Some high-risk patient populations for invasive fungal disease, such as COPD, have reduced AMP secretion, suggesting one component of increased susceptibility to infection in these populations. While AMPs are important to host immunity, limited information is available for these, especially in endemic fungal organisms.

In addition to AMPs, airway epithelia produce soluble PRRs (sPRRs) that inform host responses to invading pathogens including pentraxin 3 (PTX3), C-reactive protein (CRP), collectins (e.g., mannose binding lectin [MBL], surfactant protein A and D), serum amyloid A, and complement components (e.g., C3, C5) [94]. PTX3 is a critical regulator of antifungal host immunity in pneumocystis and aspergillosis infections [95,96,97] and may be a diagnostic tool for IPA in BAL fluid [98]. MBL, which activates complement systems, has been implicated in patients with IA, coccidioidomycosis, and *Pneumocystis* pneumonia [99,100,101]. Although many sPRRs have been implicated in human disease, limited studies have investigated the role of these in secreted receptors in human airway models. Although the mucus, AMPs, and sPRRs provide a first line of defense, fungal pathogens often adhere and directly interact with epithelial cells.

Fungal pathogens attach to the epithelium, inhibiting clearance by the host and facilitating invasion by the pathogen. Integrins on the surface of airway epithelium enable attachment of fungal spores and subsequent host response (e.g., internalization, immune cell recruitment). Integrins α3 and α5 are implicated in epithelial response to multiple fungal pathogens, including *Aspergillus*, *Rhizopus*, *Pneumocystis*, and *Paracoccidioides* [102,103]. It should be noted that these studies investigating integrins all utilized A549 cells, which may not accurately describe epithelial responses in humans. Cell wall components interacting with host integrins include the *Aspergillus* cell wall thaumatin-like protein CalA on *Aspergillus*, *Pneumocystis* glycoprotein gp120, and β-1,3-glucan [102,104]. The integrin α_3_β_1_ on epithelial cells activates epidermal growth factor receptor (EGFR) enabling invasion of *Rhizopus* [105], and inhibition with the EGFR inhibitor gefitinib blocked *Rhizopus* invasion in vitro [106]. In addition to the α3 and α5 integrins, the complement receptor CR3 has been identified as a β2 integrin that can recognize and promote conidia internalization through elevated phosphatidic acid intracellularly [107]. Integrin binding and activation lead to endocytosis and subsequent invasion in immortalized cells. While integrins play a role in host responses, other receptors on the respiratory epithelium likely contribute as well.

Understanding the receptors expressed by respiratory epithelial cells that recognize invasive fungal pathogens is critical. In immune cells, activation of PRRs (e.g., C-type lectin receptors [CLRs], Toll-like receptors [TLRs], NOD-like receptors [NLRs], and Rig-I-like receptors) mediate host responses to fungi. Dectin-1, TLRs (TLR2, TLR4, TLR7, TLR9), and NOD3 have been well studied in immune cells in response to fungi and are also expressed in human airway epithelium [108]. While these fungal sensing receptors have been reported in respiratory epithelium, little to no studies have been published to better understand these signaling pathways in a human in vitro model.

Many PRRs recognize fungal cell wall components. Cell wall chitin on *A. fumigatus* swollen conidia and hyphae binds to fibrinogen C domain-containing protein 1 (FIBCD1), a PRR apically detected on ciliated epithelial cells in human lung tissue, which suppresses mucin production, IL-8 secretion, and transcription of inflammatory mediators [109]. This may be a mechanism to avoid hyperinflammation by the host or fungal-mediated dampening of the host’s inflammatory response. Interestingly, the cell line HBE upregulates Dectin-1, a critical CLR known to activate in response to β-glucan on the fungal cell wall following infection with *A. fumigatus* [110]. Furthermore, human lung tissue immunohistochemical staining unveiled Dectin-1 on the apical bronchiolar and alveolar epithelium [111]. Dectin-1 mediates phagocytosis, suggesting that this receptor may enable conidial phagocytosis. Additionally, studies investigating the ability to kill internalized fungal organisms differ drastically, with some showing high fungal killing and others demonstrating inhibition of germination within epithelial cells.

Some reports indicate that airway epithelium phagocytose invading fungi, while others fail to observe this uptake, suggesting that phagocytosis machinery may differ between cell lines. Indeed, cell lines (i.e., A549, 16HBE, 1HAE, BEAS-2B) has been reported to internalize approximately 30–50% of *Aspergillus* conidia encountered [112,113,114]. Furthermore, *C. neoformans* adheres to and can be internalized by A549 and BEAS-2B cells [115,116]. Cigarette smoke extract increases adhesion and stimulates fungal growth but reduces the internalization of *Cryptococcus* in BEAS-2B cells, enabling growth in the airways [117]. While *Aspergillus* can be internalized, the epithelial cell often does not kill the fungus. One study observed that a failure of phagosome maturation coupled with a fusion of the fungal-containing phagolysosome with the plasma membrane contributed to a lack of killing and the ability of the fungi to continue growing within the host [118]. On the contrary, primary cultures have reported rare or no internalization of conidia by respiratory epithelial cells [119,120]. Rather than phagocytosis, one investigation demonstrated that *Aspergillus* hyphae form an actin tunnel in primary hAECs without impacting viability [119]. This observation suggests that internalization of conidia is not required for invasion. Further studies are warranted to determine if phagocytosis occurs in primary human cells and the subtype(s) responsible. Given the differences in internalization of fungal spores between cell lines and primary cells, confirmatory analyses should be conducted in primary epithelium. The immortalization process and loss of cellular diversity in human cell lines may explain these contradictory results. Furthermore, the differences in culture methods (e.g., media, ALI) may also contribute to the varied results reported. PRRs triggered by fungal pathogens on airway epithelium lead to inflammatory signaling and immune cell recruitment, highlighting the essential role of the lung epithelium in antifungal host defense.

#### 3.1.1. Communication Is Key to a Strong and Coordinated Immune Response

Secreted proteins and molecules, termed the secretome, trigger surrounding cells to respond to their environment. In the lung epithelium, the secretome can contribute to mucociliary clearance, pro-inflammatory cascades that recruit local and circulating immune cells to the site of invasion, or anti-inflammatory responses. Secretome analyses in BEAS-2B cells demonstrated that *Aspergillus* triggered thioredoxin activation, lysosomal degranulation, and cathepsin, suggesting immune defense systems within these cells [121]. Further studies identified the production of critical inflammatory and antimicrobial peptides following infection by fungal pathogens, including IL-6, IL-8, granulocyte-macrophage colony-stimulating factor (GM-CSF), human β-defensin (HBD)2, HBD9, chemokine ligand (CXCL)1, monocyte chemoattractant protein-1 (MCP-1), and TNFα, with IL-6 and IL-8 being the most widely studied [103,110,116,117,122].

*C. neoformans* capsular component glucuronoxylomannan (GXM) triggers CD14 receptor-mediated IL-8 secretion [123,124]. An acapsular strain of *C. neoformans* induced greater secretion of IL-8 as well as cellular damage in primary human bronchial epithelial cells when compared to its encapsulated parent strain [116]. Unlike the primary cells, SV40-immortalized BEAS-2B cells only produced IL-8 in response to acapsular *C. neoformans* simulations, demonstrating that some inflammatory responses may be lost in immortalized or oncogenic cell lines through the immortalizing process or lack of cell diversity. Secretion of IL-8 in response to direct contact with *Paracoccidioides* is mediated through TLR2 and the integrin α3 interaction [125], although the ligand responsible for activation has not been reported. Another study observed that protease-activated receptor antagonism blunted IL-6 and IL-8 cytokine production in *Paracoccidioides* infection of A549 cells [126]. In the setting of *Aspergillus* infection, Dectin-1 and CR3 mediated release of IL-8 and MCP-1 through induction of NFκB [127]. The pro-inflammatory signal transducer and activator of transcription 3 (STAT3)-mediated IL-6 stimulation and anti-inflammatory response (e.g., chemokines C-C motif chemokine ligand 2 [CCL2]/extracellular signal regulated kinase ½ [ERK1/2] and IL-10) have been demonstrated in cryptococcal infections in vitro in combination with either cigarette smoke or *Dermatophagoides pteronyssinus* (a trigger for allergic asthma) [117,128]. While many of these studies provide a window into the human lung, they primarily utilize immortalized and oncologic cell lines and need to be validated in primary ALI systems. Once these inflammatory factors are secreted by the epithelium, they can target circulating immune cells and recruit them to the site of infection.

Neutrophils are first responders to invading pathogens. Early neutrophil responses inhibit conidial germination and cause damage to *A. fumigatus* hyphae [129]. Many studies demonstrate the ability of the airway epithelium to release chemotactic molecules to recruit neutrophils to fungal invaders, including IL-6 and IL-8. However, these are principally in cell lines rather than primary models [130]. Utilizing an inverted ALI model, neutrophil recruitment across the epithelium can be measured in response to pathogens. Apical stimulation of primary hAECs with *A. fumigatus* lacking melanin, but not wild-type conidia, promoted transepithelial migration of human neutrophils, suggesting an immunomodulatory role for melanin [77]. However, the factor(s) responsible for this response have not been reported. Secretion of chemotactic cytokines by immortalized and primary nasal epithelial cells coincides with conidial germination, which leads to the shedding of the rodlet and melanin layers [130]. Leveraging this inverted ALI model to measure neutrophil recruitment can provide insight into initial responses by the host and identify novel therapeutic targets. While this system has been used to study *Aspergillus* infections in healthy hAECs, this can be expanded to include basal cells and neutrophils isolated from high-risk patient populations and to examine other pulmonary fungal invaders.

Macrophages, one of the most studied innate immune cells in response to fungal infections, exist as local immune cells (alveolar macrophages) and circulating immune cells recruited to the site of infection (monocytes and macrophages). Unfortunately, limited investigations report the interaction between human airway epithelium and monocytes or macrophages in response to invading fungi. Secretion of endothelin-2 has been implicated in chemoattraction and activation of macrophages. Upregulation of endothelin-2 has been noted in *Pneumocystis*-stimulated A549 cells [131]. Further investigations are needed to determine if these and other cytokines and chemokines truly play a role in epithelial-mediated host defense.

Dendritic cells (DCs) are professional antigen-presenting cells (APCs) that bridge the innate and adaptive responses to pathogens. Changes in gene expression in A549 cells drastically increased in the presence of myeloid DCs or monocyte-derived DCs (moDCs) following infection with *A. fumigatus* [132]. Genes upregulated in epithelial cells in co-cultures included PRRs (i.e., PTX3), chemokines (i.e., CCL4, CCL13, CCL18, CCL20, CXCL1, CXCL2, CXCL3, CXCL5, CXCR4), and cytokines (i.e., IL-1β, IL-8), while epithelial cells alone upregulated only superoxide dismutase 1 (SOD1). In co-cultures with A549 cells, moDCs in the alveolar compartment promoted a pro-inflammatory environment and ameliorated Mucorales invasion, whereas A549 alone secreted primarily IL-6 and IL-8 but did not restrain invasion [133]. MoDCs in the lungs are critical for shaping pro-inflammatory responses, phagocytosis of fungal pathogens, and priming of Th1 responses. CD103^+^ DCs mediate protective Th17 responses in IPA [134].

In addition to DCs, innate lymphoid cells (ILCs) enable connection between the innate and adaptive arms of immunity. ILCs are a family of lymphocytes comprising the innate counterparts of T cells. As a significant portion of resident immune cells in the lungs, they secrete cytokines, respond swiftly to pathogenic tissue damage, and shape subsequent adaptive immunity [135]. ILCs, particularly type 2 ILCs (ILC2), have been implicated in the host defense against invasive fungal pathogens [136,137]. Interestingly, deficiency of ILC2s skewed to a type 1 immunity phenotype in pulmonary cryptococcosis, which reduced fungal burden and prolonged survival [136]. The airway epithelium recruit ILC2 through the release of alarmins, such as IL-33, IL-25, thymic stromal lymphopoietin (TSLP), and high mobility group box 1 (HMGB1). Recruitment of ILC2 to the site of infection enables a Th2 response through the release of IL-4, IL-5, and IL-13 by the ILC2. Alarmins responsible for ILC recruitment have been implicated in invasive pulmonary fungal infections. Indeed, IL-33 induction from primarily alveolar type 2 epithelial cells recruits ILCs to the lung in *Cryptococcus* pulmonary infection [138,139]. IL-33 in BAL fluid samples have been detected in patients infected with *H. capsulatum* and *Pneumocystis* [140]. Furthermore, a single nucleotide polymorphism that increased the risk of developing IPA correlates with reduced serum levels of TSLP [141]. The role of ILCs in *Aspergillus* infections is primarily in the context of allergic disease, but these cells may have a protective role in invasive disease. Understanding these ILCs in humans and how they shape innate and adaptive immunity may enable insights into invasive pulmonary fungal infections and therapeutic targets.

T cell immunity is also critical in response to fungal pathogens, as demonstrated by the observation that mice lacking Th1 CD4^+^ T cells are more susceptible to IPA [142]. In primary human bronchial epithelial models, *A. fumigatus* triggers PAR-2, mediating the inhibition of IFN-β through the JAK-STAT1 pathway and skewing towards a Th2 phenotype [89,143]. This contributes to worsened outcomes in allergic inflammatory diseases. CXCL12 and its chemokine receptor, CXCR4, are critical for lymphocyte development and function. Gene analysis of A549 cells following infection with *Pneumocystis* revealed upregulation of CXCL12 and chemokine receptor (CXCR)4 [131], suggesting that respiratory epithelium assists in shaping adaptive immunity to fungal pathogens. Similar to neutrophils, macrophages, and DCs, the role of hAECs in shaping adaptive immunity remains a critical gap in knowledge. Further exploration into these responses may provide valuable targets for vaccination, especially for endemic fungi that infect healthy and immunocompromised individuals.

While some headway has been made on the role of human airway epithelium, most of these studies focus on the healthy epithelium. As noted in Table 1, severe fungal disease occurs in individuals with underlying diseases, including lung disease, other infections (e.g., influenza, HIV, SARS-CoV-2), immunomodulatory medications, and immunocompromised conditions. The contributions of the epithelium in humans susceptible to these diseases are unexplained. Understanding the mechanisms of disease in these patient populations may enable the development of novel therapeutic and preventative strategies. Transcriptional profiling during SARS-CoV-2 superinfection with IPA demonstrated that type III interferon genes and pro-inflammatory chemokines were drastically upregulated [144]. Enhanced inflammation due to increased neutrophil and monocyte recruitment to the airways in SARS-CoV-2 superinfection may drive more severe pathology, amplifying mortality in these patients. Additional studies are needed to understand the pathophysiology leading to fungal superinfections.

#### 3.1.2. DIY Local Immunity—A Novel Paradigm for Lung Immune Responses

The recent identification of M cells in the lungs may clarify how airway epithelium orchestrates complex immune responses involving epithelial barrier function and innate and adaptive immune cells [42,43]. M cells in the gut present atop classic Peyer’s patches of gut-associated lymphoid tissues (GALT), are epithelial cells with the ability to directly take up luminal antigens through transcytosis and provide their cargo to APCs localized to the basolateral side of the M cell [145,146]. DCs subsequently process and present these antigens to T cells in the lymphoid follicles resulting in B cell activation and maturation, a pathway relevant to vaccine-induced immunity. Thus, gut M cells are essential actors in initiating mucosal immunity to pathogens. While gut M cells participate in immune surveillance, they can also be co-opted and used by pathogens as portals of entry for infection [44,146,147,148]. Indeed, viruses (e.g., influenza, polio), bacteria (e.g., *Listeria*, *Mycobacterium*, *Streptococcus*), and fungi (e.g., *Candida albicans*) specifically leverage M cells to cross through the gut epithelial barrier [146,148].

Having only recently been found in the lungs of mice, the cellular origins of lung M cells and their functions in lung immunity remain unknown [42,43]. Furthermore, uninfected murine airways typically lack structured BALT. However, they are induced upon lung inflammation and infection. BALT have been identified in various diseases including fungal infections like *Cryptococcus*, *Histoplasma*, and *Pneumocystis* [149,150,151] and have shown to play a critical role in controlling influenza infections [152,153]. Inducible (i) BALT requires Th2 and Th17 immunity and occurs in a CXCL13-dependent manner in response to *Pneumocystis* infection [154]. It is unknown whether lung M cells and iBALT associate as functional immune units to regulate integrated local mucosal anti-fungal immunity. Furthermore, airway M cells and BALT function in humans is unknown. Within the airways, M cells have only been implicated as portals for infection in the case of *Mycobacterium tuberculosis* infections [44,147]. Further explorations with primary hAEC models should explore whether M cells exist in humans and, if so, what role M cells play in fungal infection.

### 3.2. Insights from Non-Fungal Pulmonary Infection Models

Invading fungal organisms are drastically understudied compared to their bacterial and viral counterparts. While epithelial and immune cells may respond differently to different types of pathogens, insights can be gained from bacterial and viral infections of the human epithelium. Here, we discuss some models of bacterial and viral infections in primary human airway epithelium with immune cells to dissect immune responses. These models can be adapted to understand the mechanisms underpinning host responses to fungal invaders. An example of this can be demonstrated with the neutrophil transmigration assay, which was described in *Pseudomonas aeruginosa* infections [155] and more recently modified to investigate IPA [77]. Placement of neutrophils in the basolateral compartment in ALI cultures better models in vivo responses compared to co-culture in submerged media. This system not only enables examination of neutrophil recruitment but also neutrophil-mediated damage to the epithelial barrier and impact on pathogen clearance as demonstrated with respiratory syncytial virus (RSV) infection [156]. A similar migration assay using peripheral blood mononuclear cells in *Bordetella pertussis* infection utilizes a flow cytometry readout to examine the migration of NK cells and macrophages [157].

Epithelial co-cultures with macrophages or DCs are limited in fungal infections. Resident alveolar macrophages enhance host defense against invading microbes. The absence of these cells in vitro removes the complex cellular interplay that likely occurs in the human lungs. An inflammation model has reported co-culture with primary hAECs and alveolar macrophages at ALI. While complicated, the addition of airway smooth muscle cells with the epithelium and alveolar macrophages strengthens the translation to the human system [158]. Another study on RSV infection introduced a dual and triple co-culture system with primary ALI bronchial epithelium, monocyte-derived macrophages (apical), and moDCs (basolateral) [159]. Interestingly, the apical presence of macrophages enabled infection of the basal moDCs following epithelial infection, but the presence of epithelium alone did not transmit RSV to moDCs. These data highlight the need to study multiple cell types in an in vitro infection model. This model also would be strengthened using alveolar macrophages rather than isolating macrophages from blood. In co-culture systems such as these, it is critical to match different cell types from the same patient, especially if isolated from an individual with an underlying condition (e.g., CF, COPD). Studies leveraging these systems in pulmonary fungal infections may enable the identification of diverse cellular networks at the site of infection prior to the recruitment of circulating immune cells.

## 4. Conclusions

The human airway epithelium is essential for host defense against invading microbes, including fungi (Figure 1). Novel techniques enable expanded culturing of primary human airway epithelial cells, recapitulating the diverse phenotype of lung epithelium in tissue. While the epithelium is vital to host response, many studies focus on immune cells rather than epithelial cells. Furthermore, studies focusing on the epithelium often utilize monomorphic cell lines that lack the cellular diversity of the epithelium. Advanced studies are warranted to expand our understanding of the response elements shaped by the epithelium in response to pulmonary fungal pathogens. Additionally, co-culture models with immune cells should be more widely used to dissect the influence of cellular cross talk on molecular mechanisms underpinning proper host responses needed to clear the infection. Together, these studies would provide new targets for vaccine development and therapeutic targets, improving morbidity and mortality associated with devastating invasive pulmonary diseases.

## Figures and Tables

**Figure 1 jof-09-00040-f001:**
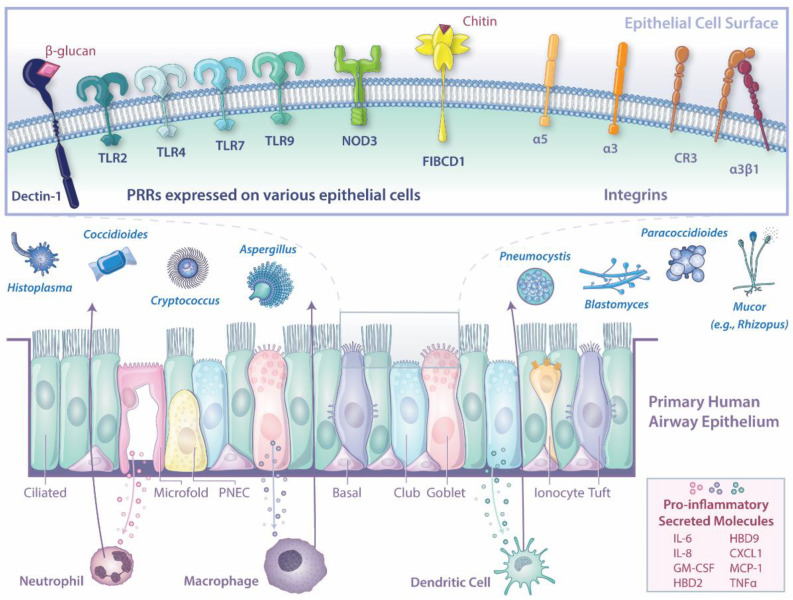
Graphical representation of the known responses of human airway epithelium to invading fungal pathogens.

**Table 1 jof-09-00040-t001:** Common pulmonary fungal infections. ABPA = allergic bronchopulmonary aspergillosis; CF = cystic fibrosis; CPA = chronic pulmonary aspergillosis; COPD = chronic obtrusive pulmonary disease; IPA = invasive pulmonary aspergillosis; PCP = *Pneumocystis* pneumonia; CM = coccidioidomycosis; PCM = paracoccidioidomycosis.

Disease	Fungal Species (**Most Common)	High-Risk Populations for Severe Disease	Global Prevalence	Mortality Rates	Ref
ABPA *	*Aspergillus fumigatus*	Asthma	4.8 million	10–30%	[3,4]
		CF			
CPA *	*A. fumigatus* ** *A. niger*, *A. flavus*, *A. nidulans*, *A. terreus*	Chronic lung disease (COPD, CF, asthma), prior mycobacterial infection, ABPA	3 million	7–80%	[5,6]
IPA *	*A. fumigatus* ** *A. flavus*, *A. niger*, *A. terreus*	Neutropenia, hematopoietic stem-cell or organ transplant, critically ill, chronic corticosteroid treatment, AIDS	250,000	40–90%	[7,8]
Acute Community- Acquired Pneumonia *	*A. fumigatus*	Prior viral infection (influenza, SARS-CoV-2), COPD	1–34% of severe viral infections	~50%	[9,10]
Pulmonary Cryptococcosis *	*Cryptococcus neoformans*, *C. gattii*	HIV, organ transplantation, malignancies, immuno-suppressive therapy, diabetes	223,000	20–55%	[11]
PJP (previously PCP) *	*Pneumocystis jirovecii*	HIV/AIDS, immune- suppressive medications (e.g., corticosteroids)	500,000	10–50%	[12,13]
Pulmonary Mucormycosis *	*Rhizopus* spp., *Mucor* spp., *Lichtheimia* spp.	Cancer, diabetes mellitus, hematopoietic stem cell or organ transplant	>10,000	38–82%	[14]
Blastomycosis ^#^	*Blastomyces dermatitidis*; *B. gilchristii*	HIV, organ transplantation, immunomodulatory therapy	1–40 cases per 100,000 (in endemic areas)	50%	[15]
CM/Valley Fever ^#^	*Coccidioides immitis* *C. posadasii*	HIV, organ transplantation, immunosuppressive therapy, diabetes, pregnancy, older age	20,000	1–88%	[16]
Histoplasmosis ^#^	*Histoplasma capsulatum*	HIV, organ transplantation, immunosuppressive therapy, infancy, older age	100,000	5–8%	[17]
PCM ^#^	*Paracoccidioides brasiliensis*; *P. lutzii*	Men; individuals in South America	15,000	6–10%	[18,19]

* Opportunistic infection; ^#^ Endemic infection.

## Data Availability

Not applicable.

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
