# Peer review of "Human Airway Epithelium Responses to Invasive Fungal Infections: A Critical Partner in Innate Immunity"

_jof, 2022, doi:10.3390/jof9010040_

Round 1

Reviewer 1 Report

To Author:

The provided manuscript is well written and covers multiple aspects of epithelial-fungal dymanics. The inclusion of differential observations based on the in vitro analysis (primary vs cell lines) used is also a significant contribution. However, there are several issues that could be addressed to improve the review and its utility for the reader.

1)    A more detailed discussion of how fungal – mold and yeast - products can promote inflammation. Specific examples to consider are secreted protease and proteins (candidalysin). Several groups have published on airway and gut, which might provide additional depth to the descriptions.

2)    Epithelial cells are essential for resistance to infection as a barrier and as a source of antimicrobial proteins. More attention should be given to the production of antimicrobial proteins in response to the fungi and inflammatory signals.

3)    Epithelial cells are an essential source of allarmins, which promote the development of ILCs. Although there are no definitive in vitro studies that have shown epithelial cells promoting ILC activity, the connection has been extensively studied in parts (i.e. epithelial cell production of alarmins, alarmins are essential for ILC production, ILC are critical regulators of early immune response and control of infection). This manuscript seems like the ideal place to synthesize the clear and concise description of how important epithelial cells are in this process.

4)    Although the focus is on human epithelial cell responses, parallels to mouse biology are real and observations in mice are relevant. Expansion to include murine studies would significantly improve the impact of the review, with acknowledgement of the species being noted.

5)    The host immune responses describe could be improved by improving the text to make a more fluid story of epithelial-mediated, fungal-driven inflammation. To this reviewer, it currently feels more like each paragraph is a free standing abstract and not a cohesive section.

Author Response

  • A more detailed discussion of how fungal – mold and yeast - products can promote inflammation. Specific examples to consider are secreted protease and proteins (candidalysin). Several groups have published on airway and gut, which might provide additional depth to the descriptions.

Response: We thank the reviewer for bringing these important secreted factors to our attention. We have expanded the review to include a discussion about lung proteases in allergic and asthma-related Aspergillus infections (lines 264-279). Furthermore, we highlighted the gap in knowledge of these factors in invasive pulmonary fungal infections. We elected not to include candidalysin in our discussion since we focused our review on infections in the lungs. Candida spp. are not pulmonary pathogens. While we discussed the gut epithelium in the context of microfold cells, this was to provide context to the extremely limited studies of microfold cells in the lungs. Thus, gut proteases and cytolytic peptides were not discussed. We kept focus for the remainder of the review on airway epithelium.

  • Epithelial cells are essential for resistance to infection as a barrier and as a source of antimicrobial proteins. More attention should be given to the production of antimicrobial proteins in response to the fungi and inflammatory signals.

Response: We updated the manuscript accordingly. Given the limited amount of information in fungal infections, we expanded the discussion of antimicrobial proteins in airway epithelium in both fungal (line 280-297) infections.

  • Epithelial cells are an essential source of allarmins, which promote the development of ILCs. Although there are no definitive in vitro studies that have shown epithelial cells promoting ILC activity, the connection has been extensively studied in parts (i.e. epithelial cell production of alarmins, alarmins are essential for ILC production, ILC are critical regulators of early immune response and control of infection). This manuscript seems like the ideal place to synthesize the clear and concise description of how important epithelial cells are in this process.

Response: We agree that ILCs are important cells to discuss in this review. As such, we have expanded our section focusing on the role of the airway epithelium in shaping immune cells responses to include ILCs (line 443-463).

  • Although the focus is on human epithelial cell responses, parallels to mouse biology are real and observations in mice are relevant. Expansion to include murine studies would significantly improve the impact of the review, with acknowledgement of the species being noted.

Response: While we agree that mouse models have provided meaningful results of the host responses in fungal infections, we elected to keep the manuscript focused on human biology. We made this decision for three main reasons: [1] expansion of the current review to discuss murine models would lead to an excessively long review and warrants its own review article; [2] results from murine models often do not translate into human biology; and [3] this review was submitted for a special issue of Journal of Fungi entitled “Immunity to Human Fungal Pathogens”. For example, we mention the promising results of the FSK vaccine that was extremely promising in mice against coccidioidomycosis but failed in clinical trials (line 135-137). We acknowledged the importance of mouse models at the beginning of section 2.1 (lines 130-144). A review focusing on comparisons of mouse and human biology is warranted in the future.

  • The host immune responses describe could be improved by improving the text to make a more fluid story of epithelial-mediated, fungal-driven inflammation. To this reviewer, it currently feels more like each paragraph is a free standing abstract and not a cohesive section.

Response: We reviewed the text and edited the manuscript to improve flow and transition. In the introduction, we started with the problem, briefly described the two types of pulmonary infections (opportunistic, endemic), and identified the gap in knowledge. In our second section, we began by describing the human airway epithelium and models commonly used to set the stage for immune responses. In our section for fungal pathogen host responses (3.1), we aimed to flow from initial inhalation and deposition of the pathogen, to indirect signaling from both the pathogen and lung epithelium, to adherence and recognition of the pathogen, to phagocytosis, and then downstream signaling to immune cells. We determined this flow to model the path of inhaled fungal organisms.

Reviewer 2 Report

This is an interesting review on human airway epithelial response to invasive fungal infections. I have only two minor comments: (i) the authors discuss about epithelial surface expressed patten recognition receptors (PRRs), and only about PTX3, a soluble PRR (sPRR; however, there are other sPRRs secreted by the airway epithelial cells, and they may be worth mentioning or discussing if there are any studies, and (ii) there are some abbreviations, which should be expanded when they appear in the text for the first time.

Author Response

  • The authors discuss about epithelial surface expressed patten recognition receptors (PRRs), and only about PTX3, a soluble PRR (sPRR; however, there are other sPRRs secreted by the airway epithelial cells, and they may be worth mentioning or discussing if there are any studies.

Response: We thank the reviewer for their helpful comment. Based on our extensive search of the current literature, we added more about PTX3 and MBL into the discussion of PRRs (lines 298-308).

  • There are some abbreviations, which should be expanded when they appear in the text for the first time.

Response: We reviewed the manuscript accordingly to ensure abbreviations were defined upon first use.
